# Reactive Carbonyl Species as Potential Pro-Oxidant Factors Involved in Lichen Planus Pathogenesis

**DOI:** 10.3390/metabo9100213

**Published:** 2019-10-03

**Authors:** Madalina Irina Mitran, Ilinca Nicolae, Mircea Tampa, Cristina Iulia Mitran, Constantin Caruntu, Maria Isabela Sarbu, Corina Daniela Ene, Clara Matei, Simona Roxana Georgescu, Mircea Ioan Popa

**Affiliations:** 1Department of Microbiology, “Carol Davila” University of Medicine and Pharmacy, 050474 Bucharest, Romania; irina.mitran@drd.umfcd.ro (M.I.M.); cristina.mitran@drd.umfcd.ro (C.I.M.); mircea.ioan.popa@gmail.com (M.I.P.); 2“Cantacuzino” National Medico-Military Institute for Research and Development, 011233 Bucharest, Romania; 3“Victor Babes” Clinical Hospital for Infectious Diseases, 030303 Bucharest, Romania; ilincanicolae84@gmail.com; 4Department of Dermatology, “Carol Davila” University of Medicine and Pharmacy, 050474 Bucharest, Romania; maria.sarbu@umfcd.ro (M.I.S.); dermatology.cm@gmail.com (C.M.); 5Department of Physiology, “Carol Davila” University of Medicine and Pharmacy, 050474 Bucharest, Romania; 6“Prof. N. Paulescu” National Institute of Diabetes, Nutrition and Metabolic Diseases, 011233 Bucharest, Romania; 7”Carol Davila” Nephrology Hospital, 010731 Bucharest; Romania; corina.daniela.ene85@gmail.com

**Keywords:** reactive metabolic intermediates, oxidative stress, antioxidants, lichen planus

## Abstract

The constant generation of reactive carbonyl species (RCSs) by lipid peroxidation during aerobic metabolism denotes their involvement in cell homeostasis. Skin represents the largest organ of the body that is exposed to lipid peroxidation. Previous studies have suggested the involvement of oxidative stress in the development of lichen planus (LP), a chronic inflammatory skin condition with a complex pathogenesis. The aim of our study is to investigate a panel of pro-oxidants (4-hydroxy-nonenal (4-HNE), thiobarbituric acid reactive substances (TBARS), and malondialdehyde (MDA)), the total antioxidant status (TAS), and thiol-disulfide homeostasis parameters (TDHP), including total thiol (TT), native thiol (NT), disulfides (DS), DS/NT ratio, DS/TT ratio, and NT/TT ratio. The comparative determinations of serum levels of 4-HNE, TBARS, and MDA in patients with LP (*n* = 31) and controls (*n* = 26) show significant differences between the two groups (4-HNE: 7.81 ± 1.96 µg/mL vs. 6.15 ± 1.17 µg/mL, *p* < 0.05, TBARS: 4.23 ± 0.59 µmol/L vs. 1.99 ± 0.23 µmol/L, *p* < 0.05, MDA: 32.3 ± 6.26 ng/mL vs. 21.26 ± 2.36 ng/mL). The serum levels of TAS are lower in LP patients compared to the control group (269.83 ± 42.63 µmol/L vs. 316.46 ± 28.76 µmol/L, *p* < 0.05). The serum levels of TDHP are altered in LP patients compared to controls (NT: 388.10 ± 11.32 µmol/L vs. 406.85 ± 9.32., TT: 430.23 ± 9.93 µmol/L vs. 445.88 ± 9.01 µmol/L, DS: 21.06 ± 1.76 µmol/L vs. 19.52 ± 0.77µmol/L). Furthermore, a negative association between pro-oxidants and TAS is identified (4-HNE – rho = −0.83, *p* < 0.01, TBARS – rho = −0.63, *p* < 0.01, and MDA – rho = −0.69, *p* < 0.01). Understanding the mechanisms by which bioactive aldehydes exert their biological effects on the skin could help define effective therapeutical strategies to counteract the cytotoxic effects of these reactive metabolic intermediates.

## 1. Introduction

Reactive carbonyl species (RCSs) are unstable intermediates that are virtually generated in all cells. In biological tissues, the majority of RCSs are derived from lipid peroxidation, a process initiated and propagated by reactive oxygen species (ROS); omega-6 and omega-3 polyunsaturated fatty acids (PUFAs) represent the main source. While the generation rate of RCSs is reduced during aerobic metabolism, it proportionally increases when cell antioxidant defense is altered [1,2,3,4,5,6,7,8]. Evidence shows that RCSs are derived from both endogenous and exogenous sources. RCSs detected in biological samples are produced by the oxidation of lipids, carbohydrates, and amino acids. Several mechanisms that might be involved in the synthesis of RCSs have been suggested, including (a) enzymatic oxidation, (b) ROS-independent non-enzymatic oxidation, and (c) ROS-mediated oxidation [8,9,10,11,12]. Several exogenous sources of RCSs have been described. Acrolein, crotonaldehyde, glyoxal, acetone, and formaldehyde are industrial pollutants that can easily enter the cell; other exogenous sources of RCSs are pharmaceutical chemistry, cigarette smoke, cotton, wood, coal, gasoline, or food additives [13,14,15,16,17]. Over 20 carbonyl intermediates, both RCSs and stable compounds, have been structurally and functionally characterized (Table 1).

The most studied lipid-derived RCSs can be classified as follows: Saturated monoaldehydes (ethanal, propanal, hexanal), unsaturated monoaldehydes (acrolein), dialdehydes (glyoxal, malondialdehyde (MDA)), ketoaldehydes (methylglyoxal, 4-oxo-2-nonenal), and hydroxyaldehydes (4-hydroxy-2-nonenal (4-HNE), 4-hydroxy-2-hexenal) [1,3]. Linoleic acid, gamma linolenic acid, and arachidonic acid preferentially form 4-HNE and 4-oxo-2-nonenal, docosahexaenoicacid and eicosapentaenoic acid generate 4-HHE, and all omega-6 fatty acids detected in cells generate MDA and propenal [2]. 4-HNE, acrolein, and methylglyoxal are the most abundant and toxic lipid-derived RCSs [1,2].

Recent studies have shown that RCSs may act as secondary messengers that activate or inhibit signaling/transcription pathways under physiological or pathological conditions. RCSs are highly reactive compounds due to their electrophilic nature and they easily react with the nucleophilic sites of proteins, DNA, and phosphoethanolamine, which generate advanced lipid peroxidation end products (ALEs). Therefore, RCSs play a dual role in vivo, their biological effects being both dose and time-dependent [1,2,11] (Table 2).

The skin is the largest organ in the body that is exposed to oxidative stress. An increasing number of studies have revealed that oxidative stress is involved in many cutaneous diseases, including lichen planus (LP) [4,18,19,20,21]. The aetiology of LP is still incompletely elucidated, however several risk factors have been described, including hepatitis C virus infection, drugs, and emotional stress [22,23,24,25,26]. The origin of cell damage in LP is considered to be the inflammatory infiltrate consisting of T lymphocytes, which contributes to local cytokine production. These cytokines stimulate the generation of ROS and RCSs that trigger keratinocyte apoptosis, which is a distinctive histopathological feature of LP. Oxidative stress increases the severity of the immune process in LP [27].

RCSs react covalently with cell components, which play the role of electron acceptor entities such as thiol groups in polypeptides and amino groups in proteins, nitrogenous bases, and lipids [6]. The resulting macromolecules may present structural alterations and changes of their physico-chemical effects (dysfunction or inactivation). To prevent tissue damage, RCSs are metabolized by several oxidoreductases, including aldo-ketoreductase (AKR), aldehyde-dehydrogenase (ADH), alcohol-dehydrogenase (ALDH), and glutathione-S-transferase (GST) [28,29]. In a previous study, we have demonstrated that an alteration in antioxidant defense systems may be involved in the pathogenesis of LP [30].

Under conditions of chronic inflammation and oxidative stress, cells develop their own mechanisms of protection. These mechanisms include redox systems such as -SH / -SS- (thiol/disulfide system), NADH/NAD^+^, NADPH/NADP^+^, and Trx(SH)2 / Trx(SS) (thioredoxin system) [31,32,33]. Recent studies have shown that the thiol-disulfide system can be regarded as an adaptive response of cells under conditions of chronic inflammation and oxidative stress. Thiols are nucleophilic compounds, which act as a scavenger for ROS (accelerating the formation of intra and intermolecular disulfide bonds), reactive nitrogen species (inducing nitration and nitrosylation of thiols), as well as for RCSs (promoting adduct formation). In this context, we suggest that the investigation of thiol-disulfide homeostasis (TDH) might be useful in order to find a strategy to counteract the cell oxidative stress present in LP [34].

Recently, TDH has been investigated in various skin disorders [35,36,37]. The parameters that evaluate TDH are native thiol (NT), total thiol (TT), disulfides (DS), and DS/NT, DS/TT, and NT/TT ratios. The determination of the oxidant/antioxidant status of an inflammatory disease can be used to assess its severity and monitor its progression and response to treatment [25,38,39,40]. Understanding the relationship between the metabolism of RCSs and LP pathogenesis could contribute to both the elucidation of the main mechanisms by which RCSs exert their biological effects in the skin and the implementation of effective therapeutic strategies to counteract the destructive effects of these reactive metabolic intermediates. To achieve these goals, we investigate a panel of RCSs (4-HNE, thiobarbituric acid reactive substances (TBARS), MDA) that are potentially involved in triggering/maintaining the LP lesions and antioxidant markers, including total antioxidant status (TAS) and thiol/disulfide homeostasis parameters (TDHP), a novel group of markers involved in the evaluation of antioxidant status in skin diseases.

## 2. Results

The serum levels of pro-oxidant markers, 4-HNE, TBARS, and MDA were higher in LP patients compared to the control group (4-HNE: 7.81 ± 1.96 µg/mL vs. 6.15 ± 1.17 µg/mL, *p* < 0.05, TBARS: 4.23 ± 0.59 µmol/L vs. 1.99 ± 0.23 µmol/L, *p* < 0.05, and MDA: 32.3 ± 6.26 ng/mL vs. 21.26 ± 2.36 ng/mL), corresponding to a 1.26-fold increase in 4-HNE levels, a 2.12-fold increase in TBARS levels, and a 1.51-fold increase in MDA levels, in LP patients compared to controls. These results are summarized in Table 3.

The serum TAS levels were lower in LP patients compared to the control group (269.83 ± 42.63 µmol/L vs. 316.46 ± 28.76 µmol/L, *p* < 0.05). In terms of thiol-disulfide homeostasis, the serum levels of NT, TT, and NT/TT ratios were lower in LP patients compared to controls, whereas the serum levels of DS, DS/NT, and DS/TT ratios were higher in LP patients compared to controls. These results are summarized in Table 4.

The determination of the global antioxidant status using the serum TAS level allows for the evaluation of all components in a sample; it is a method that is less expensive and faster than the individual determination of each parameter [4,41]. The level of thiols represents a reliable marker for evaluating the global antioxidant status, given that thiols represent 52.9% of total serum antioxidant capacity [33].

The analysis of the variations of serum 4-HNE levels according to the serum TAS levels demonstrated an inverse relationship between the two studied parameters in LP patients (rho = −0.83, *p* < 0.01) (Figure 1).

A negative correlation was also observed between the serum levels of TBARS and TAS (rho = −0.63, *p* < 0.01), as well as between MDA and TAS in LP patients (rho = −0.69, *p* < 0.01) (Figure 2, Figure 3). These correlations were also found in the control group (TAS-4HNE: rho = −0.71, *p* < 0.01, TAS-TBARS: rho = −60, *p* < 0.01, TAS-MDA: rho = −0.40, *p* < 0.01). There were no correlations between TDHP and pro-oxidant markers (Table 5).

## 3. Discussion

Oxidative stress is a hallmark of many skin disorders. Among the different types of biomolecules encountered in the skin, lipids are the major targets of ROS. The oxidative attack against PUFAs results in the generation of oxidized phospholipids (Ox-PLs). The skin contains a high amount of PUFAs in cell membranes [9,10,11]. Being associated with apoptosis, inflammation, angiogenesis, immune tolerance, and activation of various signaling pathways, Ox-PLs exert various effects [42,43,44]. Ox-PLs comprise a group of compounds including RCSs such as MDA and 4-HNE. Ox-PLs can act as antigens, given that oxidative modifications lead to the formation of immunogenic molecules. The resulting adducts are recognized by the host’s immune response in a hapten-specific manner. Antibodies against Ox-PLs have been detected in both animals and humans. Autoimmunity is a characteristic of LP and the data may contribute to the understanding of its pathogenesis [45]. Taking into account the involvement of Ox-PLs in modulating the immune response and inflammation, we consider that the evaluation of antibodies against Ox-PLs could be useful in cutaneous LP [46].

Moreover, the skin comes in direct contact with various pathogens, UV radiation, free radicals, and other pollutants that increase the risk of the peroxidation of PUFAs. The levels of RCSs in a biological system are influenced by several factors, including the different physicochemical properties of RCSs, the oxygen concentration, temperature, the relatively long half-life, and greater stability of RCSs compared to that of ROS, and the ability of RCSs to cross biological membranes and interact with targets located at a distance from their place of production [1,2,3,6,26]. Most studies have aimed at determining the lipid peroxidation levels in LP patients and have been particularly focused on the evaluation of representative metabolites in the saliva of patients with oral LP [18,19,39,47,48,49,50]. The increased amount of oxidative stress biomarkers in saliva is the result of salivary gland excretion and the effect of neural mechanisms [26,27,47]. However, there are few studies that have investigated the oxidative stress in cutaneous LP [51,52,53,54,55]. In our study, we have found an imbalance between the production of RCSs and endogenous antioxidant systems in LP patients. In line with this, we have observed an increase in serum levels of 4-HNE, TBARS, and MDA, and a reduction in serum TAS levels compared to the control group. Our findings are in concordance with the results reported in the literature. Increased MDA levels and decreased antioxidant defense in LP patients have also been shown in other studies [51,52,53,54]. However, serum levels of 4-HNE have not been previously determined in LP patients. There is only a study on erosive LP of the vulva, which has shown an increased amount of MDA and 4-HNE in the epidermis through immunohistochemical methods [56]. In LP, oxidative damage caused by RCSs could lead to the fragmentation of advanced peroxidation end products, resulting in autoantigens and an autoimmune process [47,57]. A link between lipid peroxidation and autoimmunity in LP has been previously demonstrated by the ability of 4-HNE-protein adducts and MDA-protein adducts to form specific autoantibodies [56,57].

The differences in serum levels of 4-HNE and MDA observed in LP patients compared to controls can be explained by the specificity of the method (the 4-HNE and MDA evaluation was based on a specific immunoenzymatic reaction, whereas the TBARS assay involves the interaction of thiobarbituric acid with a wide range of carbonyl compound reagents). [2]. Additionally, MDA was the major metabolite, accounting for 70% of all secondary products generated in the process of lipid peroxidation. Hexanal represents 15% and 4-HNE represents 5% of the total aldehydes [58,59].

In our study, the levels of TAS were significantly lower in LP patients than in controls. These results could be explained by the ability of RCSs to regulate cell signaling pathways, cell proliferation, and the adaptation to stress [2,29,60]. The adaptation of cells to increased levels of RCSs in vivo, including 4-HNE and MDA, and the antioxidant cell protection are achieved by the regulation of several oxidative stress-related transcriptional factors (NRF2, AP-1, NF-Kb, and peroxisome proliferator-activated receptor - PPAR), coupled with the activation of stress response pathways (mitogen-activated protein kinase - MAPK, epidermal growth factor receptor - EGFR / Akt, protein kinase C - PKC) [1,61]. At the same time, human cells have multiple defense systems against RCSs in vivo. RCSs are metabolized by either oxidation/reduction reactions (phase 1) or by conjugation reactions (phase 2) [51,62].

TDH plays an important role in skin protection against various agents (photobiological agents, chemical agents, contact allergens, etc.), epidermal differentiation, regulation of cell enzyme activity, detoxification of reactive substances, apoptosis, regulation of intracellular redox metabolism, etc. [63,64,65].

Our results show that the alteration of TDH implies an increased susceptibility to oxidative stress, materialized in the reduction in serum antioxidant capacity (increased consumption of TT and NT) and in the enhancement of the pro-oxidant potential of the serum (significant increase in DS levels as well as in DS/TT and DS/NT ratios). The resulting damage could be a key step in the onset and progression of LP lesions, given that some essential processes such as antioxidant defense, apoptosis, cell proliferation, detoxification, and regulation of enzymatic activities, etc. are disturbed. We have identified, in LP patients, a negative correlation between RCSs and thiols, as well as between RCSs and the NT/TT ratio, with no statistical significance. We have also found a positive correlation between RCSs and DS, as well as between RCSs and DS/NT, and DS/TT ratios, with no statistical significance. Consequently, we suggest that the presence of thiols in low concentrations in patients with LP fails to remove the lipid peroxides and to protect cell components against the destructive action of RCSs.

Kalkan et al., were first to analyze TDH in LP patients and determined significantly higher serum levels of NT and TT in LP patients compared to controls. The serum DS levels did not vary significantly between the two groups [33]. In our study, we found a higher level of DS in LP patients compared to controls. Upadhayay et al. found lower levels of serum protein thiols in patients with oral LP than in the control group [66].

The inverse relationships between the levels of RCSs and TAS observed in our study support the idea that 4-HNE, TBARS, and MDA might favor the development/progression of cutaneous LP lesions by exhausting antioxidant skin systems.

The skin contains a high amount of iron that promotes oxidative stress. In a previous study, we found that serum iron levels are low in patients with LP without hepatitis C, compared with those associated with hepatitis C [67]. Iron is involved in many biological processes. Changes in microelement levels may denote alterations in the body’s homeostasis. Chuykin et al. observed an inverse relationship between disease severity and iron levels in patients with oral LP [68].

We have revealed that MDA and TBARS are markers of oxidative stress in LP patients, in concordance with other studies [51,52,53,54]. Furthermore, we emphasize the role of 4-HNE as a serum marker of lipid peroxidation in LP patients, a role that has not been previously investigated. These aspects support the idea that the inactivation of RCSs might have a clinical benefit in patients with LP. Moreover, the alteration of TDH denotes that, in LP, there is an imbalance between pro-oxidants and antioxidants. However, further studies are needed to clarify whether these lipid peroxidation derived-aldehydes are involved in the development of LP lesions or are a consequence of this disorder.

## 4. Materials and Methods

### 4.1. Study Participants

This study was conducted over a three-year period, based on a prospective, observational analysis of 31 patients diagnosed with LP and a control group of 26 healthy subjects. The groups were homogeneous in terms of demographic and clinical characteristics. Informed consent was obtained from all the study participants. The procedures and the experiments we have done respect the ethical standards in the Helsinki Declaration, as well as national law. The study protocol was approved by the Ethics Committee of the Victor Babes Infectious and Tropical Diseases Hospital (2/18.09.2017). Patients were selected from those who attended the Clinic of Dermatology.

In this study, we enrolled otherwise healthy adults, aged 18 years or above, with an adequate nutritional status. The exclusion criteria included the following conditions that may alter the oxidative stress parameters values: Tobacco use, alcoholism, drug abuse, the use of corticosteroids, non-steroidal anti-inflammatory drugs, immunosuppressants, vitamins, and nutritive supplements and pregnancy. Only patients in the pre-therapeutic phase were recruited. All LP patients presented cutaneous lesions and five of them were associated with oral lesions. In all cases, the diagnosis was histopathologically confirmed.

### 4.2. Laboratory Determinations

The blood samples were collected from all study participants, after 12 h of fasting, using a holder-vacutainer system. Hemolyzed or lactescent samples were rejected. The samples were frozen at −80 degrees celsius. Several methods of quantification of lipid-derived RCSs are currently known [41]. 4-HNE, TBARS, and MDA, found in many biological samples (serum, plasma, urine, tissue, etc.), are the markers we selected to estimate the effects of aldehydes derived from lipid peroxides in LP patients. MDA forms a complex with thiobarbituric acid reactive substances (TBARS) that is measured using the spectrophotometric method (BS-3000M Semi-Automatic Chemistry Analyzer). The biological samples contain a wide range of compounds that react with TBARS. In practice, the MDA level represents the TBARS concentration in a sample. The absorbance of the complex was read at a wave-length of 532 nm. The results were expressed as micromol/L serum.

4-HNE was determined by the ELISA method, the competitive variant (semi-automatic Tecan analyzer). We used a micro ELISA plate. The wells were pre-coated with 4-HNE. The colorimetric evaluation of the final product was made at a wave-length of 450 nm. The results were expressed as microgram/mL serum. The concentration of 4-HNE in the samples was determined by comparing the optical density of the samples to the standard curve.

MDA was determined by the competitive ELISA method (semi-automatic Tecan analyzer). We used a micro ELISA plate. The wells were pre-coated with MDA. The colorimetric evaluation of the final product was made at a wave-length of 450 nm. The results were expressed as nanogram/mL serum. The concentration of MDA in the samples was determined by comparing the optical density of the samples to the standard curve.

TAS was quantified through the spectrophotometric method (HumaStar 300 analyser). The results were expressed as millimol/L serum. The determination of the global antioxidant status using the serum TAS level offers the advantage of evaluating all components in a sample, in contrast with the individual determination of each parameter, which is time consuming and expensive. However, measuring the antioxidants individually in the serum of patients with a cutaneous disorder offers the advantage of defining the causal relationships between pro and antioxidants [4,41].

TDHPs were measured using a recently developed spectrophotometric method [69]. The dynamic and reducible disulfide bonds were transformed into free functional thiol groups by using sodium borohydride (NaBH_4_, 10 mM). In the next step, the amount of NaBH_4_, which was not used in the reaction, was removed with formaldehyde (10 mM, pH 8.2). The levels of NT and TT were determined using 5,5’-dithiobis-2-nitrobenzoic acid (DTNB, 10mM). The final product, 2-nitro-5-thiobenzoate (TNB) ionized at alkaline pH and turned yellow. An automatic biochemistry analyzer (HumaStar 300, GmbH, Wiesbadencity, Germany) was used. The method allows for the evaluation of functional disulfide bonds in a sample. Half of the difference between TT and NT was considered as the DS level. DS/NT, DS/TT, and NT/TT ratios were calculated. TDHPs were represented by:NT (-SH), determined by spectrophotometric method, expressed as μmol/L serum;TT (-SH + -S-S-), determined by spectrophotometric method, expressed as μmol/L serum;DS (-S-S), determined by spectrophotometric method, expressed as μmol/L serum;DS/NT (-S-S- × 100 / -SH) was calculated;DS/TT (-S-S- × 100 / -SH + -S-S-) was calculated;NT/TT (-SH × 100 / -SH + -S-S-) was calculated.

### 4.3. Statistical Analysis

The comparison of data between groups was performed using t-tests. The relationship between pairs of two parameters was assessed by Spearman’s correlation coefficient after adequate assessment of the normality of data using the Kolmogorov–Smirnov test. We chose a significance level (*p*) of 0.05 (5%) and a confidence interval of 95% for hypothesis testing.

## 5. Conclusions

Our results suggest that increased production of RCSs and the impairment of the antioxidant defense are involved in LP pathogenesis. Of the many lipid peroxidation products, 4-HNE, TBARS, and MDA are representative of potential oxidizing factors involved in the pathogenesis of LP. We have found elevated serum levels of pro-oxidants and a reduction in serum levels of antioxidants in LP patients compared to controls. These results support the concept that 4-HNE, TBARS, and MDA might be involved in the development of LP lesions by exceeding the tissue antioxidant systems. We suggest that, in LP, the mechanism of detoxification of lipid peroxides formed in the extracellular space is related to the alteration of TDH. Further research is needed to identify specific pathways for the metabolic regulation of the biological effects of RCSs in the skin.

## Figures and Tables

**Figure 1 metabolites-09-00213-f001:**
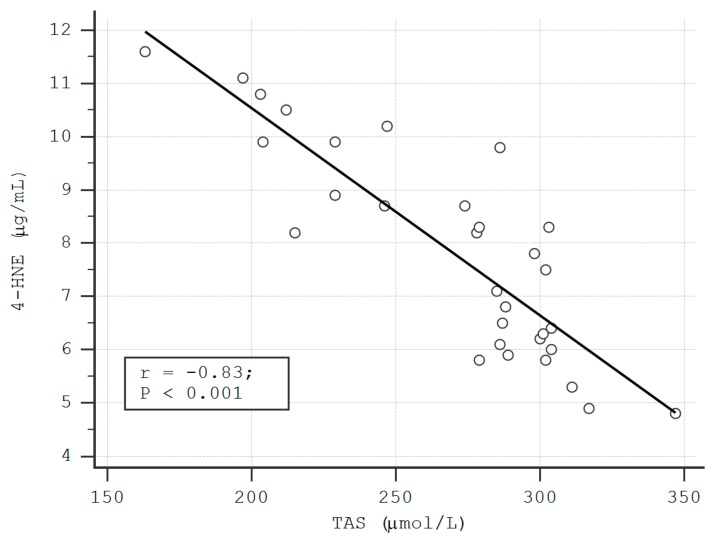
The correlation between the serum levels of 4- HNE and total antioxidant status (TAS) in LP patients.

**Figure 2 metabolites-09-00213-f002:**
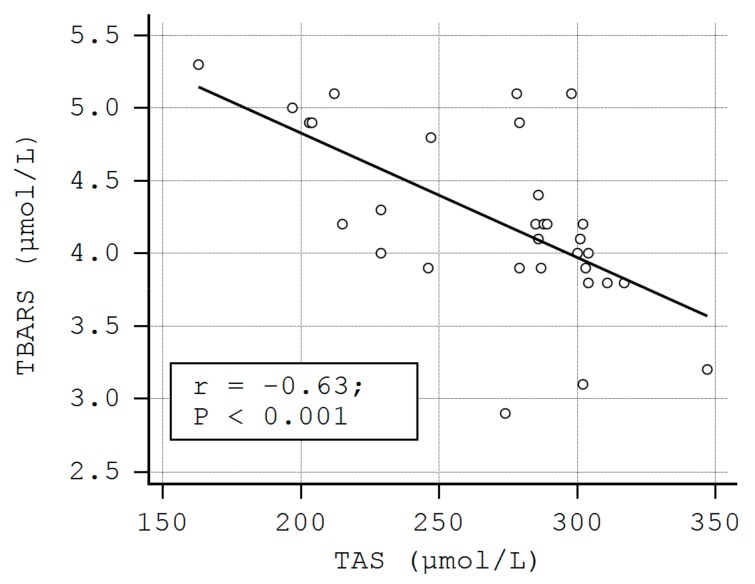
The correlation between the serum levels of thiobarbituric acid reactive substances (TBARS) and TAS in LP patients.

**Figure 3 metabolites-09-00213-f003:**
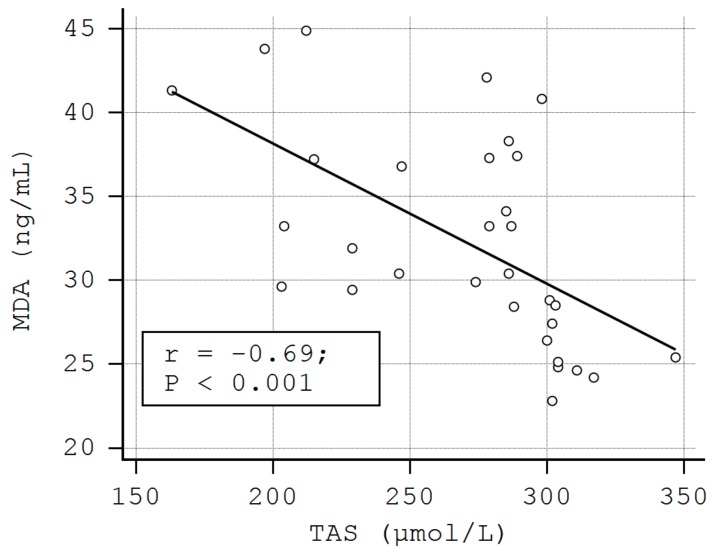
The correlation between the serum levels of malondialdehyde (MDA) and TAS in LP patients.

**Table 1 metabolites-09-00213-t001:** Lipid hydroperoxides [1,3].

RCSs	Stable Compounds
Saturated Monoaldehydes: ethanal, propanal, hexanal	Ketones: acetone, butanone;
Unsaturated Aldehydes: acrolein	Alkanes: hexane, heptane, cyclobutane
Dicarbonyls: malondialdehyde, glyoxal, methylglyoxal, isolevuglandine, 4-oxo-2-nonenal;	-
Hydroxydialdehydes: 4-hydroxy-2-nonenal, 4-hydroxy-2-hexenal:	-
Oxidized phospholipids: 1-palmitoyl-2-(5′-oxo-valeroyl)-sn-glycero-3-phosphocholine, 1-palmitoyl- 2-epoxyisoprostane-sn-glycero-3-phosphorylcholine.	-

**Table 2 metabolites-09-00213-t002:** The biological effects of Reactive carbonyl species (RCSs) derived from lipids [1,2,11].

Favourable Effects	Toxic Effects
Modulate the Signalling Pathways	Alter the cell signalling pathways
Modulate the Cell Proliferation Act as Cytotoxic Agents against Pathogens	Generate ALEs Produce cell dysfunction

**Table 3 metabolites-09-00213-t003:** The levels of pro-oxidant markers in lichen planus (LP) patients versus controls (expressed as mean and standard deviation).

Parameter	LP Patients	Controls	*p* Value
4-HNE (µg/mL)	7.81 ± 1.96	6.15 ± 1.17	<0.05 *
TBARS (µmol/L) MDA (ng/mL)	4.23 ± 0.59 32.3 ± 6.26	1.99 ± 0.23 21.26 ± 2.36	<0.05 * <0.05 *

* statistically significant.

**Table 4 metabolites-09-00213-t004:** The levels of antioxidant markers in LP patients versus controls (expressed as a mean and standard deviation).

Parameter	LP Patients	Controls	*p* Value
TAS (µmol/L) NT (μmol/L)	269.83 ± 42.63 388.10 ± 11.32	316.46 ± 28.76 406.85 ± 9.32	*p* < 0.05 * *p* < 0.05 *
TT (μmol/L) DS (μmol/L) DS/NT DS/TT NT/TT	430.23 ± 9.93 21.06 ± 1.76 5.44 ± 0.58 4.90 ± 0.46 90.20 ± 0.91	445.88 ± 9.01 19.52 ± 0.77 4.80 ± 0.24 4.38 ± 0.20 91.24 ± 0.40	*p* < 0.05 * *p* < 0.05 * *p* < 0.05 * *p* < 0.05 * *p* < 0.05 *

* statistically significant.

**Table 5 metabolites-09-00213-t005:** Correlations between thiols and pro-oxidant markers.

Parameter	4-HNE		TBARS		MDA	
-	rho	*p*	rho	*p*	rho	*p*
TAS NT	−0.83 −0.11	< 0.01 * 0.54	−0.63 −0.32	< 0.01 * 0.08	−0.69 −0.22	< 0.01 * 0.22
TT	−0.11	0.54	−0.27	0.12	−0.26	0.14
DS	0.14	0.43	0.20	0.26	0.03	0.86
DS/NT	0.15	0.41	0.26	0.15	0.07	0.67
DS/TT	0.15	0.41	0.26	0.15	0.07	0.67
NT/TT	−0.15	0.41	−0.26	0.15	−0.07	0.67

* statistically significant.

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
