# Peer review of "Reactive Carbonyl Species as Potential Pro-Oxidant Factors Involved in Lichen Planus Pathogenesis"

_metabolites, 2019, doi:10.3390/metabo9100213_

Round 1

Reviewer 1 Report

Although the findings are not too surprising, the paper adds some potentially interesting aspects to the pathophysiology of lichen planus.

The TBARS-method, however,  is questionable as for many aspects. Therefore, the authors should add data of another independent method for MDA and also for total antioxidant status.

The potential role of oxidized phospholipids comprising also carbonyl species has not been addressed and should be included into the discussion (e.g. Deigner/Hermetter, Current Opinion in Lipidology, June 2008, Vol. 19)

Reviewer 2 Report

In this work, Mitran and co-authors measured serum levels of 4-HNE and MDA in patients affected by cutaneous lichen planus. Since 4-HNE and MDA are by-products of the majority RCGs, authors considered them as representative indicators of lipid peroxidation rate in patients. Authors reported a statistically significant increase of 4-HNE and MDA in patients with respect to controls. Moreover, they reported that 4-HNE and MDA values inversely correlate with TAS in patients.

The work is well presented and the manuscript clearly written.

Major concerns

As the authors pointed out, their findings are not completely novel in the field of cutaneous lichen planus research. Moreover, the study was conducted on a limited number of patients.

They should increase the relevance of the study.

A possibility is to report the serum levels of some component of TAS, such as folic acid or vitamin B12, as previously done by other authors for oral lichen planus patients.

In addition, authors might report about other parameters considered associated with lichen planus pathogenesis, such as iron status.

Minor concerns

Authors should report whether a correlation between TAS and 4-HNE (and/or MDA) levels exists in controls.

Methods description can be improved

Round 2

Reviewer 1 Report

the paper has improved after revision

Reviewer 2 Report

I have no additional comments